# Transarterial Embolization for Active Gastrointestinal Bleeding: Predictors of Early Mortality and Early Rebleeding

**DOI:** 10.3390/jpm12111856

**Published:** 2022-11-07

**Authors:** Chloé Extrat, Sylvain Grange, Alexandre Mayaud, Loïc Villeneuve, Clément Chevalier, Nicolas Williet, Bertrand Le Roy, Claire Boutet, Rémi Grange

**Affiliations:** 1Department of Radiology, University Hospital of Saint-Etienne, 42270 Saint-Priest-en-Jarez, France; 2Department of Gastro-Enterology, University Hospital of Saint-Etienne, 42270 Saint-Priest-en-Jarez, France; 3Department of Oncologic and Digestive Surgery, University Hospital of Saint-Etienne, 42270 Saint-Priest-en-Jarez, France

**Keywords:** gastrointestinal bleeding, embolization, rebleeding, mortality, lactate

## Abstract

Background: The aim of this study was to determine predictive factors of early mortality and early rebleeding (≤30 days) following transarterial embolization (TAE) for treatment of acute gastrointestinal bleeding. Methods: All consecutive patients admitted for acute gastrointestinal bleeding to the interventional radiology department in a tertiary center between January 2012 and January 2022 were included. Exclusion criteria were patients: (1) aged < 18-year-old, (2) referred to the operation room without TAE, (3) treated for hemobilia, (4) with mesenteric hematoma, (5) lost to follow-up within 30 days after the procedure. We evaluated pre and per-procedure clinical data, biological data, outcomes, and complications. Results: Sixty-eight patients were included: 55 (80.9%) experienced upper gastrointestinal bleeding and 13 (19.1%) lower gastrointestinal bleeding. Median age was 69 (61–74) years. There were 49 (72%) males. Median hemoglobin was 7.25 (6.1–8.3) g/dL. There were 30 (50%) ulcers. Coils were used in 46 (67.6%) procedures. Early mortality was 15 (22.1%) and early rebleeding was 17 (25%). In multivariate analysis, hyperlactatemia (≥2 mmol/L) were predictive of early mortality (≤30 days). A high number of red blood cells units was associated with early rebleeding. Conclusion: This study identified some predictive factors of 30-day mortality and early rebleeding following TAE. This will assist in patient selection and may help improve the management of gastrointestinal bleeding.

## 1. Introduction

Acute gastrointestinal bleeding (GIB) is a clinical situation that can lead to significant mortality and morbidity without urgent care. Lower Gastrointestinal Bleeding (LGIB) originates downstream of the Treitz ligament, while Upper Gastrointestinal Bleeding (UGIB) originates upstream of the Treitz ligament. Despite advances in endoscopic hemostasis and adjuvant pharmacologic treatment, the hospital mortality rate from UGIB remains 10% and has not significantly improved over the past 50 years [1]. Although surgery is the historical treatment for GIB, it is associated with a high risk of complications and an estimated mortality of 10–30% [2]. Immediate endoscopy is the examination of choice for the diagnosis and treatment of gastrointestinal bleeding and should not be delayed for more than 24 h following admission [3]. Rebleeding in UGIB occurs in 7–16% of cases despite endoscopic therapy [4].

In patients with clinical evidence of rebleeding following successful initial endoscopic hemostasis, the European Society of Gastroenterology recommends repeat upper endoscopy with hemostasis if indicated. In the case of failure of this second attempt at hemostasis, transarterial embolization (TAE) or surgery should be considered (strong recommendation, high-quality evidence) [5,6]. For LGIB, TAE should be reserved for the treatment of acute potentially life-threatening GIB in hemodynamically unstable patients or in patients not amenably treated by endoscopic interventions [5].

Studies on the efficacy of endovascular treatments have mainly focused on one location [7] or on a type of embolic agent [8,9] by assessing clinical success. Evaluation of prognostic factors is lacking. Additionally, few published series have analyzed factors predicting early death after TAE. In addition, studies on predictive factors of early rebleeding show contradictory results. Loffroy et al. reported that the use of coils in patients with coagulation disorders is associated with rebleeding [10]. Mohan et al. reported that patients with a history of malignancy were more likely to rebleed within 30 days and that younger patients (<60 years) were significantly less likely to experience rebleeding within 30 days [11]. Finally, the rate of rebleeding varies between the studies, from 13% [12] to 46.8% [11].

The aim of this monocentric retrospective study was to determine predictive factors of early mortality (≤30 days) and early rebleeding (≤30 days) following TAE in the treatment of acute GIB during a 10-year period.

## 2. Materials and Methods

### 2.1. Study Population

All of the patients referred to our hospital for GIB who were treated by TAE based on clinical decisions in emergency and CT scan between January 2012 and January 2022 were retrospectively reviewed. GIB was defined as intra luminal hemorrhage of the gastrointestinal tractus diagnosed by endoscopy and/or CT scan. Inclusion criteria were all patients with GIB who were treated by emergency TAE. Exclusion criteria were patients (1) aged < 18 years old, (2) referred to the operating room and who did not have TAE, (3) treated for hemobilia without associated GI lumen bleeding, (4) with isolated mesenteric hematoma, and (5) lost to follow-up within 30 days of the procedure.

### 2.2. Clinical Data

The following data were collected from electronic medical records. The patient demographics included age, gender, and comorbid conditions prior to TAE. Comorbid conditions included diabetes, coronaropathy, high blood pressure (HBP), chronic renal failure (CRF), active cancer, cancer in remission, and anticoagulation or antiplatelet treatments. Biological data included prior coagulopathy, lactate rate, transfusion requirements, and number of RBC units transfused. Imaging data included endoscopic or angiographic findings, causes of GIB, and CT findings (active bleeding, pseudoaneurysm, and location of bleeding).

Procedure data included angiographic findings, embolization material, vessel embolized, and duration of procedure. The post-procedure data included the occurrence of minor and major complications, rebleeding, type of management for rebleeding, length of hospitalization, hospital admission, length of hospitalization in an intensive care unit, and mortality.

### 2.3. Pre-Procedure

Pre-angiographic investigations sometimes involved an endoscopy performed at the onset of acute gastrointestinal bleeding. We therefore recorded the number of gastroscopies and colonoscopies performed before embolization. The endoscopy was considered positive if acute bleeding was observed. Endoscopic treatment was considered as a failure if active bleeding was not stopped.

Patients underwent an abdominal CT scan (SOMATOM DEFINITION AS 64, Siemens AG, Medical Solution, Erlangen, Germany). Patients received ≥90 mL contrast medium (Xenetix 350, Guerbet, Villepinte, France) with a flow rate ≥3 mL/s. Unenhanced and contrast-enhanced liver CT at the arterial and portal phases were performed according to the standard-of-care protocol of our hospital. A bleed was considered active when iodine contrast was present at the arterial phase and increased at the portal phase. Pseudoaneurysm was considered as a rupture of arterial caliber without an increase in the portal phase.

### 2.4. Transarterial Embolization Methods and Techniques

TAE procedures were performed by 1 of the 10 interventional radiologists whose experience ranged from 2 to 30 years after multidisciplinary consultation (surgeon, clinician, and radiologist). After local anesthesia with lidocaine, the right common femoral artery was accessed routinely. Celiac, superior mesenteric, and/or inferior mesenteric angiograms were performed to determine the focus of mesenteric injury using a 4F catheter and a hydrophilic guidewire (Terumo^®^, Tokyo, Japan). Supraselective catheterization was systematically performed using a 2.7F microcatheter (Progreat^®^, Terumo, Tokyo, Japan). TAE was performed under fluoroscopic monitoring using fibered microcoils (Interlock^®^, Boston Scientific, Marlborough, MA, USA), N-butyl-2-cyanoacrylate-NBCA (Glubran2^®^, GEM, Viareggio, Italy), gelatine sponge (Gelitaspon^®^, Gelita Medical, Amsterdam, Holland), or microparticles (Embosphere^®^ Microspheres, BioSphere Medical, Rockland) depending on the vascular wound and at the discretion of the radiologist. After the procedure, complete angiograms were performed to confirm that bleeding had been successfully controlled.

When angiography remained negative despite active bleeding in CT scans or endoscopy, “empiric” TAE of the suspected bleeding artery was performed at the discretion of the interventional radiologist. No spasmolytic agents to reduce bowel peristaltic were administrated, and no provocation test was conducted.

### 2.5. Patient Follow Up

After TAE, all the patients were closely monitored for clinical signs and symptoms suggestive of ischemic complications or recurrent bleeding until discharge or death. These clinical findings were supplemented by laboratory studies.

Patients’ long-term outcomes, specifically incidence of rebleeding, mortality, and procedure-related complications, were collected from patient charts. CT follow-up and endoscopic examination were not routine practices performed following TAE in our unit.

### 2.6. Definitions

Technical success was defined as the cessation of angiographic extravasation immediately after TAE based on angiographic findings. Clinical success was defined as resolution of signs and symptoms of bleeding during the 30-day follow-up after TAE and without required endoscopic treatment, surgery, or repeat TAE or death related to massive blood loss during this period of time. Prior coagulopathy was defined as INR > 1.5, PT < 50%, or PC < 150 G/L. Acute renal failure was defined as a rapid and reversible decline in the glomerular filtration rate. Endoscopic treatment failure was defined as failure to stop bleeding or early recurrence within 48 h of endoscopy. The number of RBDs transfused was calculated from the day of embolization to 48 h after embolization.

Rebleeding events were classified as early events if they occurred ≤30 days following TAE and as late events if they occurred >30 days following TAE. Complications were defined as per operative complications if they occurred during TAE and as post-operative complications if they occurred during follow-up. Minor and major complications were separated using the CIRSE classification [13]. Grades I and II were considered minor complications, and grades III, IV, and V were considered major complications.

### 2.7. Outcomes

The primary endpoint of our study was to identify any predictors of early death (≤30 days) after TAE.

Secondary endpoints were to identify predictive factors of early rebleeding (≤30 days) and clinical failure after TAE.

### 2.8. Statistical Analysis

Results are presented as median and inter-quartile for continuous variables and as number and frequency for categorical variables. Categorical variables were compared with the Chi-squared test, and continuous variables were compared with Student’s t-test. A univariate analysis was performed to assess the association between early mortality and predictive factors. A multivariate regression analysis was performed using the backward stepwise selection model. Variables with a *p* value < 0.1 in the univariate analysis were entered into the multivariate analysis. The odds ratio (OR) and 95% confidence intervals were reported. A statistically significant difference was considered for *p* < 0.05. Statistical analyses were performed using the R software.

### 2.9. Ethical Considerations

This study was approved by the ethics committee of our institute (CHU de Saint-Etienne “Terre d’Ethique”, IRBN112021).

## 3. Results

Between January 2012 and January 2022, 789 patients were referred for TAE in our institute. A flowchart of the patient sample population is presented in Figure 1.

### 3.1. Patient Characteristics

Sixty-eight patients admitted to our interventional department for GIB requiring TAE were included. The detailed patient characteristics are presented in Table 1. The median age was 69 (61–74) years, including 49/68 (72%) males. Regarding previous treatments, 14/68 (20.6%) patients had received antiplatelet therapy, and 19/68 (27.9%) had received anticoagulation therapy. There were 7/68 (10.3%) patients with a history of cancer, and 25/68 (36.8%) patients had an active cancer. There were 55/68 (80.9%) patients with UGIB (Figure 2), including 45 with duodenal and 10 with gastric bleeding, and 13/68 (19.1%) patients with LGIB, including six with colonic bleeding, four with jejunal bleeding, and three with rectal bleeding (Figure 3).

The clinical presentation of the patients was hematemesis in 22/68 (32.4%) patients, melena in 31/68 (45.6%) patients, and rectorragia in 27/68 (39.7%) patients. The main causes of bleeding were ulcer (50%) and post-operative (20.6%). The causes of bleeding are detailed in Table 2.

### 3.2. Pre-Procedure

Regarding pre-procedure investigations of the patients, 62/68 (91.1%) patients had a pre-operative angiographic CT: active bleeding was detected in 41/68 (60.3%) patients, and pseudoaneurysm was detected in 9/68 (13.2%) patients. In total, 40/68 (58.8%) patients had a gastroscopy, of which 38 showed active bleeding; 35 experienced failure of endoscopic treatment. On the three patients treated for rectal bleeding, two experienced failure of endoscopy. The third patient had abundant active bleeding, which pushed us to perform treatment by TAE directly. The embolized vessels were in the superior rectal artery in all three cases.

### 3.3. Procedure Data

The details of the per-procedure data are presented in Table 3. Out of the patients, 12/68 (17.6%) had no abnormalities on fluoroscopic angiogram and were treated by empiric embolization. Among the 12 patients treated by empiric TAE, seven patients had active bleeding on CT, and three patients had active bleeding on endoscopy. These elements made it possible to orientate TAE. Two patients had neither active bleeding on endoscopy nor on CT but did have a duodenal ulcer: an occlusion of the gastroduodenal artery using coils was performed.

TAE was performed using coils in 46/68 (67.6%) patients, NCBA in 8/68 (11.8%) patients, a combination of coils and resorbable gelatine in 5/68 (7.4%) patients, gelatine sponge in 3/68 (4.4%) patients, microparticles in 4/68 (5.9%) patients, and a combination of microparticles and gelatine sponge in 2/68 (2.9%) patients.

The main artery embolized was the gastroduodenal artery, which was embolized in 43/68 (63.2%) patients. The median time for the procedure was 60 (40–87) min.

### 3.4. Post-Angiography Course

Detailed clinical outcomes after TAE are presented in Table 4. Technical success was achieved in 68/68 patients. There were 7/68 (10.3%) complications per procedure, including 3/68 (4.4%) non-target embolizations, 1 (1.5%) coil migration, and 3 (4.4%) hematomas at the puncture site. There were 13/68 (19.1%) post-operative complications. The most common post-operative complication was acute renal failure in 9/68 (13.2%) patients, which did not require dialysis.

The median length of stay in intensive care units was 3 (1–6) days. The median length of hospital stay was 12 (6–24) days. The median follow-up time was 5 (1–11) months. During the follow-up period, 32/68 (47.0%) patients died, including 15/68 (22.1%) patients who died during the 30 days following the procedure. The clinical success rate was 50/68 (73%) patients, and 17/68 (25%) patients experienced early rebleeding. Among these 17 patients, 3 were treated by surgery, 3 were treated by endoscopy, 2 were treated by repeat TAE, 4 were treated by endoscopy followed by surgery, 2 were treated by repeat TAE followed by endoscopy followed by surgery, and 3 were treated by conservative treatment; 9/17 (53%) died within 30 days.

### 3.5. Predictors of Early Mortality

In the univariate analysis (Table 5), acute renal failure (OR = 3.75 CI = 1.02–13.7 *p* < 0.05), lactate ≥ 2 mmol/L (OR = 6.8 CI = 1.37–33.2 *p* < 0.01), and ulcers (OR = 3.59 CI = 1.01–12.73 *p* < 0.05) were statistically associated with early mortality. Early mortality was not predicted by age, sex, coagulopathy disorders, symptoms, location of bleeding, or any type of embolic agent.

In multivariate analysis, lactate ≥ 2 mmol/L (OR = 6.10 CI = 1.54–24.2 *p* = 0.03) was associated with early mortality.

### 3.6. Predictors of Early Rebleeding

In univariate analysis (OR = 1.19 CI = 1.04–1.38, *p* = 0.011) and multivariate analysis (OR = 1.92 CI = 1.01–1.35, *p* = 0.037), higher red blood cell (RBC) units administered in a transfusion was associated with early rebleeding. Early rebleeding was not predicted by age, sex, comorbidities, symptoms, hemostasis anomalies, source of bleeding, embolized artery, or embolic agent.

## 4. Discussion

Several authors have studied the prognosis of patients treated with TAE for gastrointestinal bleeding. However, most of these studies have focused on LGIB or UGIB [14], on a particular embolizing agent [8], or a particular clinical situation [15] or have had limited pre- or per-procedural clinical information [11,16]. Some studies include patients with arterial splanchnic bleeding without intraluminal bleeding [9]. There is a lack of a uniform reporting system because of the variety of definitions for clinical success. For example, the Society of Interventional Radiology defined clinical success as resolution of symptoms within 30 days after TAE, whereas other authors use 7 days after TAE as their time cut-off [17,18,19]. Furthermore, studies do not systematically report the number of patients lost to follow-up [20], nor do they report if these patients are excluded from the study or included in the final statistical analysis. We believe that the study of rebleeding and early mortality are the most appropriate to evaluate the effectiveness of TAE for GIB.

Causes of early death following TAE are multifactorial and include acute renal failure, infection, multiorgan failure, and bleeding. The early mortality rate (21.1%) is comparable to other previous studies [10,16]. Lactate level is known to be a useful and rapid tool for assessing severity of disease in critically ill patients [21]. It is also known as a predictor of post-operative infection, cardiopulmonary dysfunction, renal impairment, and increased mortality after elective cardiac surgery [22]; aortic dissection [23]; and liver transplantation [24]. Cellular stress, tissue hypoxemia, infection, and various critical illnesses are triggers for the accumulation of serum lactate [25]. In the critically ill patient, normal blood lactate is less than 2 mmol/L. Lactate ≥ 2 mmol/L was the only predictor of early mortality ≤30 days in the present study. To our knowledge, these biological data have never been investigated in prognostic studies of TAE for GIB. To our knowledge, no previous study has shown that hyperlactatemia is a predictor of early mortality after TAE. The advantage of these data is that they are easy to use, are quickly accessible, and represent an objective measure in patient assessment. They would allow clinicians to closely monitor patients with GIB treated by TAE with hyperlactatemia ≥ 2 mmol/L.

In the present study, the early rebleeding rate (25%) was comparable to previous studies and was approximately 33% (range 9–66%) for UGIB [26]. This is in line with other studies, such as the study of Loffroy et al. in 2009 [10], in which among the 16 patients who presented with early rebleeding, 3 (18.8%) were treated by repeat TAE, with a secondary clinical success of 77.2%. The influence of rebleeding on patient outcomes is contradictory. Mohan et al. [11] showed that rebleeding within 30 days was associated with increased odds of 30-day mortality that were more than 45 times higher than normal. On the other hand, Loffroy et al. [10] reported 28.1% (16/57 patients) mortality within 30 days of TAE, with only three deaths caused by early rebleeding.

Regarding early rebleeding, a greater amount of RBC transfusion has been found as a predictive factor. However, we did not find that the presence of coagulopathy was associated with early rebleeding, contrary to previous studies in the literature [20]. Lee et al. [27] showed that coagulopathy is associated with rebleeding. In a retrospective study of 114 patients, Mohan et al. [11] identified two predictive factors of rebleeding: age > 60 years and patients with known malignancy. A longer time to angiography, a greater number of RBCs, the use of coils as the only embolic agent, and previous surgery have also been found [28,29,30] to be predictive factors for early rebleeding in some studies.

We have shown that TAE can be performed successfully, even when angiography fails to visualize active bleeding. Indeed, among the 12 empiric TAE, only 1/12 (8.3%) patients died early from bleeding, with no statistical differences between empiric and targeted TAE in terms of early rebleeding and mortality. This result is consistent with the meta-analysis by Yu et al. [31], which found a clinical success rate of 74.7% for empiric TAE and no statistically difference between empiric and targeted TAE in terms of rebleeding. However, this result can also be explained by the potential spontaneous drying up of the bleeding, explaining the absence of active bleeding.

In our study, embolic agents were not predictive of early rebleeding or early mortality. Coils were the most common embolic agents used in this study. Several studies have demonstrated a statistically significant association between the use of coils and higher rebleeding rates [10,28,32]. In contrast, Kim et al. [33] showed a clinical success rate of 82% for UGIB TAE using NCBA. Despite this, NBCA was used less frequently than coils because NBCA can be challenging to use, especially in high-flow arteries such as the gastroduodenal artery, and it requires extensive experience to be used successfully. The results of our study temper the results of these previous studies, highlighting the influence of an embolic agent on the risk of recurrent bleeding. This study shows that no single embolic agent should be preferred in principle. However, the technical skill of all embolic agents, including NBCA, allows the operator to adapt to all clinical situations and to avoid complications such as non-target embolization or coil migration. Only two transient bowel ischemia occurred in this study sample among patients with LGIB. Neither case required surgical resection. This is in line with the systematic review of Beggs et al. [2], who did not report any ischemic complications for UGIB, which can be explained by the rich collateral blood supply of the gastro duodenal artery. In contrast, studies on LGIB showed higher rates of ischemic complications in relation to a less developed anastomotic network. For instance, Bua-Ngam et al. [34] showed an ischemic complication rate of 13%. Recently, interventional teams have even used pre-operative splanchnic or inferior mesenteric artery TAE before surgery for rectal cancer [35] or esophagus cancer [36] as pre-ischemic conditioning to prevent post-operative anastomosis leakage. In addition, our study confirms the safety profile of TAE for GIB, and no major complications were noted. The main complications were acute renal failure, the causes of which were multifactorial and related to the contrast injection and volume depletion.

This study has several limitations. First, this study was retrospective. Second, our investigation represents the experience of a single institution. Third, because the patients in this study were reviewed over a 10-year period, very minor technical differences in TAE among the primary interventional operators as well as differences in medical care during the study period could have contributed to disparities in clinical outcomes over time. Additionally, there was no systematic scanning or endoscopic follow-up after the operation to assess ischemic complications. Finally, the number of events, such as early death and early rebleeding, are relatively small to ensure the robustness of our multivariate analyses.

In conclusion, this retrospective study suggests hyperlactatemia as a potential and previously undocumented predictor for early mortality after TAE for GIB. As the blood lactate level is easily evaluated in clinical practice, these findings have practical implications for clinicians’ early assessments of GIB so that they can adjust patient management accordingly. Further prospective and larger multicenter studies are needed to support these data. In addition, large-scale studies comparing risk factors for early mortality of TAE with endoscopic and surgical treatment for GIB seem necessary.

## Figures and Tables

**Figure 1 jpm-12-01856-f001:**
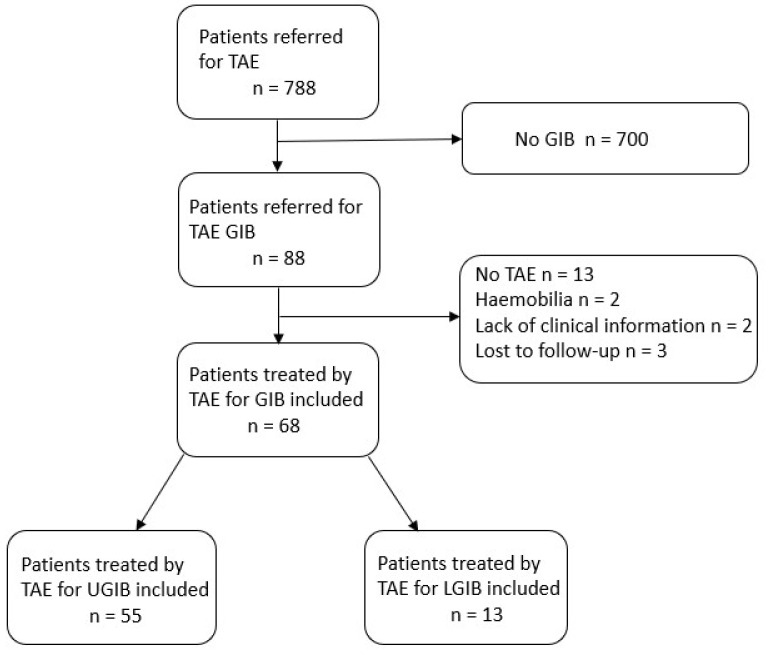
Flow chart of the study population.

**Figure 2 jpm-12-01856-f002:**
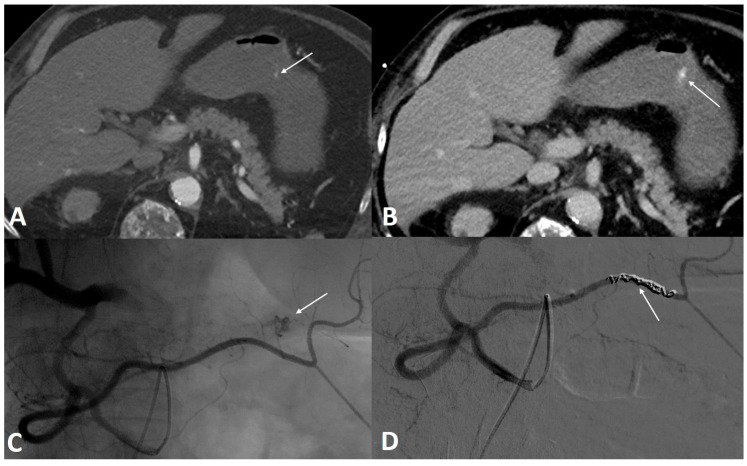
A 56-year-old male was referred to the emergency room for profuse hematemesis. Gastroscopy confirmed the presence of antral bleeding, with no possibility of stopping the bleeding. (**A**) Axial section CT scan injected at the arterial phase, confirming the active contrast extravasation. (**B**) Axial section abdominal CT scan injected at portal phase showing an increase of the leak of iodine contrast. (**C**) Angiography performed within the superior mesenteric artery using a Cobra probe showed an active contrast leak from a branch of the right gastroepiploic artery. (**D**) After embolization with three coils, the angiographic control showed a clear stop of the bleeding.

**Figure 3 jpm-12-01856-f003:**
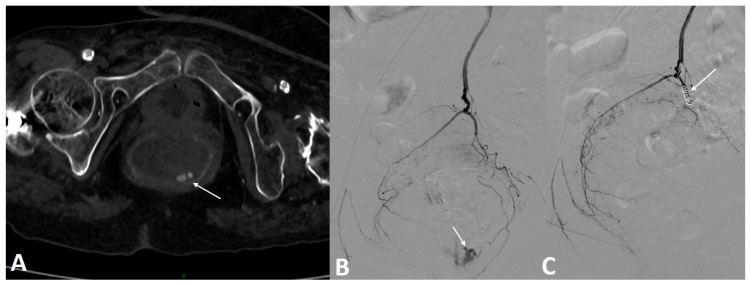
A 73-year-old patient on anticoagulation was referred for profuse rectal bleeding. Colonoscopy failed to stop the bleeding. (**A**) Axial section CT scan injected at arterial time confirming the active contrast extravasation in the rectal lumen. (**B**) Arteriography of the inferior mesenteric artery confirmed the presence of active bleeding from the left superior rectal artery. (**C**) Embolization with three fibered coils allowed a complete stop of the bleeding.

**Table 1 jpm-12-01856-t001:** Characteristics of the 68 patients included in the study.

Variable	n = 68
Age, years	69 (61–74)
**Male n (%)**	49 (72)
**Comorbidities n (%)**	
Diabetes	12 (17.6)
Coronaropathy	13 (19.1)
HBP	39 (57.3)
CRF	7 (10.3)
Cancer in remission	7 (10.3)
Active cancer	25 (36.8)
Anticoagulation therapy	19 (27.9)
Antiplatelet therapy	14 (20.6)
**Clinical presentation n (%)**	
Melena	31 (45.6)
Rectorragia	27 (39.7)
Hematemesis	22 (32.4)
**Biology n (%)**	
Lactate ≥ 2 mmol/L	25 (36.8)
Prior coagulopathy	24 (35.3)
Hb nadir	7.25 (6.1–8.3)
Acute renal failure	14 (20.5)
**Pre-operative CT n (%)**	62 (91.1)
Active bleeding	41 (60.3)
Pseudoaneurysm	9 (13.2)
**Localization of bleeding n (%)**	
UGIB	55 (80.9)
Duodenal	45 (66.2)
Gastric	10(14.7)
LGIB	13(19.1)
Colon	6 (8.8)
Jejunum	4 (5.9)
Rectum	3 (4.4)
Ileum	0
**Pre-operative gastroscopy n (%)**	40 (58.8)
Active bleeding	38 (55.8)
Failure of endoscopic treatment	35 (51.4)
**Pre-operative coloscopy**	8 (11.8)
Positive coloscopy	4 (5.9)
Failure of endoscopic treatment	4 (5.9)
**Transfusion n (%)**	62 (91.2)
RBC units	5 (3–8)

Quantitative parameters are presented as median and interquartile range (IQR, 25th–75th percentile). RBC: Red blood cell, CRF: Chronic renal failure, Hb: Hemoglobin.

**Table 2 jpm-12-01856-t002:** Causes of bleeding in the study population.

Variable	n = 68
**Ulcer**	**34 (50)**
**Post operative**	**14 (20.6)**
Duodeno pancreatectomy	7 (10.3)
Sphincterotomy	5 (7.4)
Gastrectomy	2 (2.9)
**Cancer**	**13 (19.1)**
Gastric cancer	8 (11.8)
Pancreatic cancer	4 (5.9)
Rectal Cancer	1 (1.5)
**Pancreatitis**	**2 (2.9)**
**Idiopathic**	**2 (2.9)**
**Diverticulosis**	**2 (2.9)**
**Angiodysplasia**	**1 (1.5)**

**Table 3 jpm-12-01856-t003:** Per procedure characteristics of the 68 patients.

Variable	n = 68
**Angiographic data. (%)**	
Pseudoaneurysm	11 (16.2)
Empirical Embolization	12 (17.6)
**Arteries Embolized n. (%)**	
Gastroduodenal	43 (63.2)
Upper Mesenteric	8 (11.8)
Inferior mesenteric	5 (7.4)
Left colic	2 (2.9)
Superior rectal	3 (4.4)
Left Gastric	4 (5.9)
Splenic	2 (2.9)
Gastroepiploic	2 (2.9)
Pancreaticoduodenal	1 (1.5)
Left Hepatic	1 (1.5)
Right Gastric	1 (1.5)
Right Hepatic	1 (1.5)
**Embolic Agents n. (%)**	
Coils	46 (67.6)
NCBA	8 (11.8)
Coils + Gelatine Sponge	5 (7.4)
Microparticles	4 (5.9)
Gelatine Sponge	3 (4.4)
Microparticles + gelatine sponge	2 (2.9)
**Duration of procedure (min)**	60 (40–87)

Quantitative parameters are presented as median and interquartile range (IQR, 25th–75th percentile). NBCA: N-butyl Cyanoacrylate.

**Table 4 jpm-12-01856-t004:** Outcome of the 68 patients included in the study.

Variable	n = 68
**Technical Success n (%)**	**68 (100)**
**Clinical Success n (%)**	**50 (73)**
**Mortality during follow-up n (%)**	**32 (47)**
**Day-30 mortality n (%)**	15 (22.1)
**Per-operative Complications n (%)**	**7 (10.3)**
Non-target embolization	3 (4.4)
Coil Migration	1 (1.5)
Hematoma at puncture site	3 (4.4)
**Post-Operative Complications n (%)**	**13 (19.1)**
Acute renal failure without dialysis	9 (13.2)
Bowel Ischemia	2 (2.9)
Splenic Ischemia	2 (2.9)
**Recurrence of Bleeding n (%)**	**19 (27.9)**
Early ≤ 30 days	17 (25)
Delayed > 30 days	2 (2.9)
**Management of Early Rebleeding n (%)**	**17 (25)**
Surgery	3 (4.4)
Repeat TAE	2 (2.9)
Endoscopy followed by Surgery	4 (5.9)
Endoscopy	3 (4.4)
TAE followed by endoscopy followed by surgery	2 (2.9)
Conservative treatment	3 (4.4)
**Length of hospital stay (days)**	**12 (6–24)**
**Length of stay in intensive units (days)**	**3 (1–6)**
**Duration of follow-up (months)**	**5 (1–11)**

Quantitative parameters are presented as median and interquartile range (IQR, 25th–75th percentile).

**Table 5 jpm-12-01856-t005:** Univariate and multivariate regression analysis for early death and early rebleeding.

		Early Death ≤ 30 Days				Early Rebleeding ≤ 30 Days		
	Univariate Analysis		Multivariate Analysis		Univariate Analysis		Multivariate Analysis	
Characteristics	OR	*p* Value	OR	*p* Value	OR	*p* Value	OR	*p* Value
**Demographics data**								
Age ≥ 70	0.99 (0.94–1.03)	0.64	─	─	0.71 (0.23–2.25)	0.57	─	─
Male	0.72 (0.21–2.47)	0.6	─	─	0.62 (0.19–2.04)	0.43	─	─
HBP	1.15 (0.36–3.69)	0.81	─	─	1.5 (0.48–4.69)	0.48	─	─
Diabetes	0.66 (0.13–3.41)	0.62	─	─	1 (0.24–4.22)	1	─	─
Chronic renal failure	1.48 (0.26–8.50)	0.66	─	─	1.23 (0.21–6.99)	0.82	─	─
Active cancer	0.55 (0.16–1.97)	0.36	**─**	**─**	0.28 (0.07–1.1)	0.07	─	─
**Coagulation disorder**								
Prior coagulopathy	0.38 (0.12–1.22)	0.1			0.51 (0.17–1.68)	0.25		
Curative Anticoagulation	0.31 (0.06–1.52)	0.15	─	─	0.70 (0.19–2.49)	0.58	─	─
Antiplatelet	0.20 (0.02–1.70)	0.14	─	─	1.16 (0.31–4.37)	0.81	─	─
Lactates ≥ 2mmol/L	6.3 (1.7–23.2)	**0.006**	6.10 (1.54–24.2)	**0.01**	2.76 (0.87–9.71)	0.08	─	─
**Clinical Presentation**								
Duodenal Bleeding	1.54 (0.43–5.50)	0.51	─	─	1.92 (0.55–6.78)	0.31	─	─
Hematemesis	1.41 (0.43–4.66)	0.57	─	─	0.86 (0.26–2.87)	0.8	─	─
Active infection	3.35 (0.77–14.56)	0.11			2.7 (0.63–11.58)	0.18		
Acute renal failure	3.75 (1.02–13.7)	**0.043**	─	─	1.94 (0.55–6.91)	0.3	─	─
**Pre-operative data**								
Active bleeding on CT	2.11 (0.59–7.48)	0.25	─	─	0.92 (0.30–2.81)	0.89	─	─
Failed pre-operative gastroscopy	1.56 (0.49–5.00)	0.46	─	─	1.08 (0.36–3.25)	0.89	─	─
**Hb and transfusion requirement**								
Hb < 8	2.9 (0.89–9.48)	0.07	─	─	2.86 (0.93–8.83)	0.07		─
Number of RBC	1.07 (0.94–1.2)	0.31	─	─	1.19 (1.04–1.38)	**0.011**	1.17 (1.01–1.35)	**0.037**
**Causes and arteries embolized**								
Ulcer	3.59 (1.01–12.73)	**0.048**	─	─	3.16 (0.97–10.3)	0.056	─	─
Gastroduodenal Artery	0.35 (0.09–1.40)	0.14	─	─	2.74 (0.75–5.48)	0.91	─	─
**Procedure**								
Empiric Embolization	0.66 (0.13–3.41)	0.62	─	─	1 (0.23–4.2)	1	─	─
Embolization with Coils only	0.29 (0.06–1.48)	0.138	─	─	0.42 (0.11–1.68)	0.22	─	─

HBP: High Blood Pressure Hb: Hemoglobin.

## Data Availability

The data presented in this study are available on request from the corresponding author.

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
