# Peer review of "Transarterial Embolization for Active Gastrointestinal Bleeding: Predictors of Early Mortality and Early Rebleeding"

_jpm, 2022, doi:10.3390/jpm12111856_

Round 1
Reviewer 1 Report
Thank you for giving me a chance to review this article. TAE for GIB is one of the most important strategy for rescue, however, there are som points remained. Please check, revise, and answer for these points.
1. Table 3, duplications of letters occurs, I cannot read.
2. Figure 1, arrows are not cleanly connected to next boxes.
3. Multivariate analyses, if pssible, please recalculate by stepwise method.
4. Elevtion of lactate is usually critical indicator in patients with hemorrahgic shock; therefore, results are very natural. From these background, not surprisingly and may not be a new findings.
Reviewer 2 Report
Dear authors,
Thank you for submitting your manuscript jpm-1885491 for consideration in the JPM as an original research article. It is a retrospective observational study from the field of interventional radiology aiming to define predisposing factors for an unfavorable clinical prognosis after endovascular therapy of gastrointestinal bleeding (GIB). The results conclude that increased lactate is associated with increased mortality, and high numbers of transfused RBC-units is associated with rebleeding risk.
C001_The main comment of the reviewer is that the results are not novel. The fact that increased lactate in blood is associated with mortality is textbook clinical knowledge and not specific to the endovascularly treated GIB. Moreover, an increased number of RBC-unit causes transfusion coagulopathy, also known to be associated with rebleeding risk.
C002_The results should be put in a clinical context and provide a perspective. An interesting approach would compare endovascular to endoscopic therapy and surgical therapy risk factors.
C003_ It would be interesting if the authors could elaborate on how their results affect their clinical routine.
C004_ the manuscript will benefit from professional English proof editing with proficiency in medical terminology
Author Response
Reviewer 2
Dear authors,
Thank you for submitting your manuscript jpm-1885491 for consideration in the JPM as an original research article. It is a retrospective observational study from the field of interventional radiology aiming to define predisposing factors for an unfavorable clinical prognosis after endovascular therapy of gastrointestinal bleeding (GIB). The results conclude that increased lactate is associated with increased mortality, and high numbers of transfused RBC-units is associated with rebleeding risk.
C001_The main comment of the reviewer is that the results are not novel. The fact that increased lactate in blood is associated with mortality is textbook clinical knowledge and not specific to the endovascularly treated GIB. Moreover, an increased number of RBC-unit causes transfusion coagulopathy, also known to be associated with rebleeding risk.
Reply C001_: Thank you for your valuable comments. Indeed, these results are not surprising as it is widely used in post-operative cardiac surgery. However, to our knowledge, no study has investigated lactate as a prognostic biomarker in patients treated by TAE. To our knowledge, this element has not been investigated in studies of haemostasis TAE. The predictive factors are rather oriented towards the embolic agent, used for example. However, we believe that biomarkers are of great relevance for assessing predictive factors of early mortality.
Indeed, the increase in RBC units is a known predictive factor for rebleeding. However, some studies on TAE show that use of coils is associated with a risk of recurrent bleeding. The fact that no embolic agent was associated with recurrence of bleeding in the present study seems to be an important finding. It tempers the results of some earlier studies, showing an increased risk of recurrence with coils (https://doi.org/10.1016/S1051-0443(07)61825-9; https://doi.org/10.1016/j.cgh.2009.02.003) , or gelatin sponge (https://doi.org/10.1148/radiology.177.1.2399325).
Accordingly, we have added the following text to the manuscript (line 340):
“The results of our study temper the results of these previous studies, highlighting the influence of an embolic agent on the risk of recurrent bleeding. This study shows that no single embolic agent should be preferred in principle. However, the technical skill of all embolic agents, including NBCA, allows the operator to adapt to all clinical situations and to avoid complications such as non-target embolization or coil migration.”
C002_The results should be put in a clinical context and provide a perspective. An interesting approach would compare endovascular to endoscopic therapy and surgical therapy risk factors.
Reply C002_: Thank you for your suggestion. We agree that it would be an interesting study to carry out. This element has been added in the revised manuscript.
“In addition, large-scale studies comparing risk factors for early mortality of TAE with endoscopic and surgical treatment for GIB seem necessary.”
C003_ It would be interesting if the authors could elaborate on how their results affect their clinical routine.
Reply C003_: Thank you for this comment. We have not emphasized enough that lactate testing could be an additional element in predicting prognosis and thus anticipating management. This has been added in the revised manuscript:
They would allow clinicians to closely monitor patients with GIB treated by TAE with hyperlactatemia ≥2 mmol/l.
We also emphasised the importance of tempering studies associating the use of certain embolic agents with recurrent bleeding. The technical skill of all embolic agents, on the other hand, makes it possible to adapt to the clinical situation and arterial anatomy to improve the performance of the treatment and to avoid complications related to the embolic agent. The following sentences have been added to the revised manuscript:
“The results of our study temper the results of these previous studies, highlighting the influence of an embolic agent on the risk of recurrent bleeding. This study shows that no single embolic agent should be preferred in principle. However, the technical skill of all embolic agents, including NBCA, allows the operator to adapt to all clinical situations and to avoid complications such as non-target embolization or coil migration.”
C004_ the manuscript will benefit from professional English proof editing with proficiency in medical terminology
A English proofreading of the original manuscript has been completed.
Reviewer 3 Report
I would like the authors to elaborate on these items. Why was TAE used in rectal bleeding and which vessel was embolized? Why was empiric TAE performed? Were these high risk patients with a tendency to rebleed ? What criteria were used to determine that a combined approach was preferable?
Author Response
Reviewer 3
Thanks very much for taking your time to review this manuscript. We appreciate your constructive comments and suggestions. Please find my itemized responses in below and my revisions/corrections in the re-submitted files.
I would like the authors to elaborate on these items.
1.Why was TAE used in rectal bleeding and which vessel was embolized?
Reply 1 In 2/3 cases it was a failure of the endoscopic treatment. In the other 1/3 cases, it was an ectasic artery and abundant active bleeding which pushed us to directly undergo treatment by TAE. The embolized vessels were the superior rectal artery in all 3 cases. These various elements have been added to the revised manuscript
This has been added on the revised manuscript:
“On the three patients treated for rectal bleeding, two experienced failure of endoscopy. The third patient had abundant active bleeding, which pushed us to perform treatment by TAE directly.. The embolised vessels were in the superior rectal artery in all three cases”
Variable |
n=68 |
Angiographic data. (%) |
|
Pseudoaneurysm |
11(16.2) |
Empirical Embolization |
12(17.6) |
Arteries Embolized n. (%) |
|
Gastroduodenal |
43(63.2) |
Upper Mesenteric |
8(11.8) |
Inferior mesenteric |
5(7.4) |
Left colic |
2(2.9) |
Superior rectal |
3(4.4) |
Left Gastric |
4(5.9) |
Splenic |
2(2.9) |
Gastroepiploic |
2(2.9) |
Pancreaticoduodenal |
1(1.5) |
Left Hepatic |
1(1.5) |
Right Gastric |
1(1.5) |
Right Hepatic |
1(1.5) |
Embolic Agents n. (%) |
|
Coils |
46(67.6) |
NCBA |
8(11.8) |
Coils + Gelatine Sponge |
5(7.4) |
Microparticles |
4(5.9) |
Gelatine Sponge |
3(4.4) |
Microparticle + gelatine sponge |
2(2.9) |
Duration of procedure (min) |
60(40-87) |
|
|
2.Why was empiric TAE performed? Were these high risk patients with a tendency to rebleed ?
Reply 2 : : Of the 12 patients treated by empiric TAE, 7 patients had active bleeding as shown by CT, 3 patients had active bleeding as shown by endoscopic exam. These elements made it possible to orientate the TAE. Two patients had neither active bleeding on endoscopy nor on CT but a duodenal ulcer. In these cases, an occlusion of the gastroduodenal artery using coils was performed.
These elements have been added to the manuscript to clarify:
“Among the 12 patients treated by empiric TAE, 7 patients had active bleeding on CT, and 3 patients had active bleeding on endoscopy. These elements made it possible to orientate TAE. Two patients had neither active bleeding on endoscopy nor on CT but did have a duodenal ulcer: an occlusion of the gastroduodenal artery using coils was performed.”
- What criteria were used to determine that a combined approach was preferable?
Reply 3 : Thank you for this comment.
Concerning the combined treatments used for the treatment of rebleeding, it is more a reflection of the failure of one of the modalities which brings about the change in the type of treatment. Indeed, the use of "+" is confusing in Table 4. We have therefore modified the table 4 by using the term "followed by".
Variable |
n=68 |
Technical Success n (%) |
68(100) |
Clinical Success n (%) |
50(73) |
Mortality during follow-up n (%) |
32(47) |
Day-30 mortality n (%) |
15(22.1) |
Per-operative Complications n (%) |
7(10.3) |
Non-target embolization |
3(4.4) |
Coil Migration |
1(1.5) |
Haematoma at puncture site |
3(4.4) |
Post-Operative Complications n (%) |
13(19.1) |
Acute renal failure without dialysis |
9(13.2) |
Bowel Ischemia |
2(2.9) |
Splenic Ischaemia |
2(2.9) |
Recurrence of Bleeding n (%) |
19(27.9) |
Early ≤ 30days |
17(25) |
Delayed>30 days |
2(2.9) |
Management of Early Rebleeding n (%) |
17(25) |
Surgery |
3(4.4) |
Repeat TAE |
2(2.9) |
Endoscopy followed by Surgery |
4(5.9) |
Endoscopy |
3(4.4) |
TAE followed by endoscopy followed by surgery |
2(2.9) |
Conservative treatment |
3(4.4) |
Length of hospital stay (days) |
12(6-24) |
Length of stay in intensive units (days) |
3(1-6) |
Duration of follow-up (months) |
5(1-11) |
Table 4: Outcome of the 68 patients included in the study
Quantitative parameters are presented as median and interquartile range (IQR, 25th-75th percentile).
This has been modified in the revised manuscript :
Among these 17 patients, 3 were treated by surgery, 3 were treated by endoscopy, 2 were treated by repeat TAE, 4 were treated by endoscopy followed by surgery, 2 were treated by repeat TAE followed by endoscopy followed by surgery and 3 were treated by conservative treatment; 9/17(53%) died within 30 days.
Round 2
Reviewer 1 Report
The authors have revised along with my suggestions.
Reviewer 2 Report
Dear Editor,
Dear authors,
In the resubmitted manuscript I am respectfully lacking the scientific hypothesis and the novelty.
Hyperlactemia as a mortality predictor in gut ischemia is a piece of basic medical knowledge.